# Effect of Hot Deformation Process Parameters on Microstructure and Corrosion Behavior of 35CrMoV Steel

**DOI:** 10.3390/ma12091455

**Published:** 2019-05-06

**Authors:** Qiumei Yang, Yajun Zhou, Zheng Li, Daheng Mao

**Affiliations:** 1School of Mechanical and Electrical Engineering, Central South University, Changsha 410083, China; YangQMei141@126.com (Q.Y.); lizheng2016@csu.edu.cn (Z.L.); mdh@csu.edu.cn (D.M.); 2National Key Laboratory of High-Performance Complex Manufacturing, Central South University, Changsha 410083, China

**Keywords:** thermal deformation parameters, 35CrMoV steel, grain size, electrochemical corrosion

## Abstract

Hot deformation experiments of as-cast 35CrMoV steel, with strain rates of 0.01 s^−1^ and 10 s^−1^, deformation temperatures of 850, 950, and 1050 °C, and an extreme deformation reaching 50%, were carried out using a Gleeble-3810 thermal simulator. Electrochemical corrosion experiments were conducted on the deformed specimens. The microstructure was observed by optical microscope (OM), and the corrosion morphology and corrosion products of the specimens were investigated by scanning electron microscopy (SEM), energy disperse spectroscopy (EDS), confocal laser scanning microscopy (CLSM), and X-ray diffraction (XRD) techniques. The results show that the grain size increased gradually with an increase in the deformation temperature at the same strain rate, whereas the corrosion resistance deteriorated. At the same deformation temperature, the grain size becomes smaller as the strain rate increases, which enhances the corrosion resistance. This is mainly attributed to the fine grains, which can form more grain boundaries, increase the grain boundary area, and accelerate the formation of the inner rust layer at the beginning of corrosion. Moreover, fine grains can also refine the rust particles and enhance the bonding strength between the inner rust layer and the matrix. The denseness and stability of the inner rust layer increases as the corrosion process progresses, thereby improving corrosion resistance.

## 1. Introduction

It is well-known that 35CrMoV steel is a high-quality steel with good hardenability, creep properties, high strength, and a high fatigue limit. It is generally used to manufacture gear, marine power equipment, advanced turbine blowers and impellers, compressor engines, and other important parts which operate under high stress [1]. Under certain conditions, such as high stress and complex marine conditions (especially for complex marine conditions with strong corrosiveness), Cl^−^ has a strong penetrating power that can cause pitting corrosion of carbon steel and easy penetration of the surface defects. Furthermore, the invasion of Cl^−^ can force the surface of the passivation film to rupture, which will aggravate the stress corrosion cracking failure of the steel [2,3,4]. Therefore, in addition to the mechanical properties, its corrosion performance also seriously threatens the service life, safety, and efficiency of the marine components.

During thermal processing, different thermal deformation parameters have significant effects on the microstructure and properties of the material [5,6]. Numerous studies have been done into the evolution of the microstructure during thermal deformation in recent years [7,8,9]. Kingkam et al. [10] investigated the effects of the deformation temperature and strain rate on the dynamic recrystallization behavior of high-strength low-alloy steel. It was observed that an increase of deformation temperature led to an increase of grain size and the effects of the microstructure on corrosion behavior were briefly discussed. Xiao et al. [11] observed that the smallest dynamic recrystallization grains were obtained at a higher strain rate and lower deformation temperature during hot deformation. Moreover, high temperatures and low strain rates are beneficial for grain growth. Lin et al. and Xu et al. established the constitutive equation and process diagram of 25Cr3Mo3Nb steel. It is reported that the change of grain size is closely related to the strain rate and, when the strain rate is 1 s^−1^, a uniform distribution and fine grains can be obtained [12]. Additionally, Huang [13] studied the dynamic recrystallization of 35CrMo steel and found that it was affected considerably by the deformation temperature and strain rate and was difficult to recrystallize at a lower temperature and higher strain rate. Lin et al. [14] studied the microstructure of a Ni-Fe-Cr-based superalloy by thermal compression experiments and the experimental results were consistent with the established dynamic recrystallization model. Quan et al. [15] obtained fine-grained, equiaxed-crystal re-crystallized structures of ASS alloys generated by the thermal deformation activation of dynamic recrystallization. These studies explored the evolution of the microstructure during the thermal deformation process, but rarely involved changes in microstructure that affected the corrosion resistance.

Some studies have exhibited that grain refinement can significantly improve the corrosion resistance of the material [16]. For instance, Argade and Alvarez-Lopez [17,18] showed that grain boundaries can serve as corrosion barriers and considered the role they play in delayed corrosion. Balyanov et al. believed that the rapid passivation of ultrafine Ti makes ultrafine crystals, which have a stronger corrosion resistance than coarse-grained industrial pure Ti [19]. Aung [20] discovered that small grains can reduce the corrosion rate of the AZ31B magnesium alloy. Zhao discussed the effect of the microstructure on the corrosion performance of the AZ91 alloy, and pointed out that the presence of a second phase and surface film can act as a corrosion barrier to prevent corrosion of the matrix [21,22,23]. Schino explored the inter-granular corrosion rate of AISI 304 austenitic stainless-steel and found that it decreases with a decrease of grain size in an H_2_SO_4_-FeSO_4_ solution. This is due to the increase of grain boundary per unit-volume and the decrease of Cr consumption caused by carbide precipitation, as the inter-granular corrosion rate goes down with the increase of the grain boundary area [24]. The results of Pradhan et al. showed that a fine-grained microstructure has similar or better corrosion resistance to a coarse-grained structure under the optimum distribution of grain boundary characteristics [25]. In addition, it was suggested that the grain boundary is favorable for passivation, high-density dislocations, and grain boundaries, and that grain refinements are propitious to form passivation film as grain refinement is helpful for the rapid formation of a passivation film on the surface, which can reduce the corrosion rate [26]. It was considered that coarse crystal samples are more susceptible to sensitization and inter-granular corrosion than fine-grained samples because their anode and cathode area are lower [27]. It was reported that the degree of sensitization decreases exponentially with the increase of the grain boundary surface-area [28,29,30], which means that Cr is diffused from the inside to the grain boundary, thereby reducing the growth of carbide precipitates, delaying the sensitization process, and improving corrosion resistance. The above studies show that the grain size, in both magnesium alloy and stainless-steel, affects the corrosion behavior of the material and that grain refinement can improve the corrosion resistance of the material. However, research on the influence of the microstructure evolution of 35CrMoV steel on the corrosion performance in the process of thermal deformation needs further study. Therefore, by adjusting the thermal processing parameters to study the changes of the microstructure in the thermal deformation process, and then analyzing the influence of grain size on corrosion behavior, more appropriate thermal deformation process parameters are obtained, which can provide technical guidance for obtaining large-size forgings with a fine grain size and good performance.

In this study, as-cast 35CrMoV is taken as the research object and thermal compression experiments are carried out at different strain rates and temperatures. The microstructure of 35CrMoV steel after thermal deformation is characterized by optical microscope (OM) and the grain sizes of specimens with different thermal deformation parameters are obtained by a professional micro-image analysis system. The corrosion behaviors of different grain-size specimens obtained under different thermal deformation process parameters are studied in a 3.5 wt% NaCl solution through a three-electrode system. Furthermore, the corrosion performance of the samples is evaluated by electrochemical impedance spectroscopy (EIS) and TAFEL curves, and the corrosion morphology are observed by the energy disperse spectroscopy (EDS), confocal laser scanning microscopy (CLSM), and Hyperfield 3D microscopic systems. Therefore, we can determine the influence of grain size on corrosion resistance and reveal the influence mechanism of the microstructure, obtained under different parameters, on corrosion resistance.

## 2. Materials and Methods

### 2.1. Materials

Our research material, 35CrMoV steel, is a typical low-alloy steel for industrial use and its chemical composition is shown in Table 1. The as-cast 35CrMoV steel specimens of dimensions Φ 400 × 800 mm and hot compression specimens with size Φ 10 × 15 mm were cut at 1/2R (R is the radius of the as-cast, 200 mm) by a wire cutting machine. Then, the thermal compression test was carried out by using the Gleeble-3810 thermal simulator. The graphite sheet was placed between the two ends of the specimens, as well as the contact surface of the pressure head, before the beginning of the experiment, and lubricant was applied onto the two ends of the specimens so that the specimens could be subjected to uniform deformation during heating and compression. Before the beginning of the experiment, the graphite sheet was placed between the two ends of the specimens and the contact surface of the indenter, and lubricant was applied onto the two ends of the specimens to keep the temperature uniformly distributed during heating and compression. The process of thermal compression is shown in Figure 1. First, it was heated to 1200 °C, at a heating rate of 10 °C/s^−1^, for 5 min and then cooled to the deformation temperature (850 °C, 950 °C, 1150 °C), at a cooling rate of 10 °C/s^−1^, with heat preservation for 1 min to eliminate the temperature gradient of the specimens. Then, an extreme deformation, reaching 50%, was carried out at a strain rate of 0.01–10 s^−1^. Finally, the specimens were quenched in water immediately after compression and the same heat treatment was carried out, the specimens were heated to 850 °C for 30 min, then oil-cooled, re-heated to 620 °C for 30 min, and finally, were water-cooled.

### 2.2. Microscopic Analysis

The microstructure of the specimens was observed by an Olympus DSX500 optical microscope, and the average grain size was calculated by an artificial cut-off method, according to the standard of GB/T6394-2002. Before the metallographic experiment, the specimens, with a diameter of 10 × 5 mm^2^, were taken along the axial direction of the compressed specimens by a wire cutter, and then they were ground, from coarse to fine sandpaper, on a grinding machine and polished with diamond spray until the surface of the specimens was bright. On this basis, the specimens were corroded in a self-made corrosion solution (2.5 g picric acid + 50 mL distilled water + 2 g dodecyl benzenesulfonic acid, sodium salt), heated, and kept at 60–80 °C for 4–8 min. After the corrosion was finished, the specimens were removed from the surface with cotton, washed with distilled water, and dried with hot air, so that the microstructure of the specimens could be observed under an optical microscope.

### 2.3. Electrochemical Measurements

Three dynamic potential polarization experiments were performed on the electrochemical workstation (Chi660e, CH Hua, Shanghai, China) using a three-electrode corrosion measurement system. Before the experiment, the specimens were ground to 1500# on the grinder and the polishing cloth was mechanically polished with diamond spray polish until the surface was smooth and free from scratches. Finally, it was washed with distilled water, ethanol, and then dried with hot air. The potentiodynamic polarization was measured by a three-electrode system in a 600 mL corrosion medium cell containing 3.5 wt% NaCl with a saturated calomel (Ag/AgCl) electrode as a reference electrode and platinum-plate as an auxiliary electrode. The surface area of the specimens exposed to the solution was 10 × 5 mm^2^. The specimens were immersed in the corrosive medium for 2400 s to obtain a stable open-circuit potential and, on this basis, electrochemical impedance spectroscopy (EIS) was carried out with a frequency range of 0.01–10^5^ Hz and an AC excitation signal amplitude of 10 mV, which can obtain the phase angle diagram of Nyquist and Bode. Then, the appropriate equivalent circuit model was selected and the EIS results were found by equivalent-fitting with the ZSimpWin software. Ultimately, the potentiodynamic polarization curve was measured in the range of −1 to 0 V vs. SCE, at a scanning rate of 0.5 mVS^−1^, and the corrosion potential (E_corr_), corrosion current density (I_corr_), anode slope (B_a_), and cathodic slope (B_c_) were obtained by Tafel fitting. 

### 2.4. Corrosion Surface Analysis

In order to better observe the corrosion morphology of the specimen surface and analyze the elements of the corrosion products, the surface of the specimens was first dried with a compressed air gun and then was observed by scanning electron microscope (SEM) (EvoMA 10C Zeissi Jena, Oberkochen, Germany). Moreover, the chemical elements of the surface corrosion products were analyzed by energy disperse spectroscopy (EDS) and the surface morphology of corrosion specimens is obtained by using Hyperfield 3D microscopic systems (Keyence VHX-5000, Osaka, Japan) and confocal laser scanning microscopy (CLSM) (Zeiss Axio LSM700, Oberkochen, Germany).

## 3. Results and Discussion

### 3.1. Microstructure

The microstructure, after hot deformation at different deformation temperatures and strain rates, is shown in Figure 2. The deformation temperatures and strain rates have a significant effect on the microstructure of the 35CrMoV steel. There is a dynamic recrystallization structure in all of the samples and the grain size distribution is not uniform, as displayed in Figure 2. A statistical analysis of the grain size in Figure 2 was carried out and the statistical results are shown in Figure 3. At the strain rate of 0.01 s^−1^, the grain size increased gradually from 12.5 to 15.4 μm when the deformation temperatures increased from 850 to 1050 °C (increasing by 20%), which is consistent with the law at the strain rates of 10 and 0.01 s^−1^. The results show that, at the same strain rate, the grain growth with the deformation temperature quickly increases, resulting in coarse grains. This can be attributed to higher deformation temperature making the dislocation motion more intense and more favorable to grain boundary migration, which enhances the diffusion and dislocation slip of vacancy atoms, increasing the nucleation of re-crystallization and the dynamic re-crystallization rate. As a result, small grains are continuously swallowed by large grains and the microstructure is coarsened. When the strain rates are increased from 0.01 to 10 s^−1^ at the deformation temperature of 850 °C, the grain sizes decrease gradually from 12.5 to 10.7 μm, decreasing by 16%. It can be noticed that, as the strain rates are increased from 0.01 to 10 s^−1^ at the deformation temperatures of 950 and 1050 °C, the grain change is consistent with that at 850 °C, which indicates that an increase in strain rate is conducive to grain refinement and acquiring a fine-grained structure at the same deformation temperature. This is mainly due to larger strain rates accelerating the deformation of the samples, and so more strain storage-energy is generated, making the dynamic re-crystallization nucleation rate activate [31]. Thus, there are more phase deformation nuclei and a higher nucleation rate. Moreover, the re-crystallized grain does not have enough time to grow in this case, which reduces the dynamic recovery rate. In addition, the higher strain rate leads to enhanced accumulated strain energy and dislocation density in grains, which makes the dynamic re-crystallization easier to nucleate. It can also be seen from Figure 2 that the number of re-crystallized grains formed along the grain boundary at a strain rate of 10 s^−1^ is higher than that at a strain rate of 0.01 s^−1^. Furthermore, the dynamic recrystallization rate is lower at lower-deformation temperatures, which can be attributed to the decrease of the grain boundary mobility at low deformation temperatures and high strain rates, as well as a tendency toward incomplete dynamic re-crystallization.

### 3.2. Electrochemical Analysis

#### 3.2.1. Potentiodynamic Polarization Curve

In this section, the effects of grain sizes, obtained under different process parameters, on the corrosion resistance of 35CrMoV steel are discussed by potentiodynamic polarization experiments. Moreover, the electrochemical impedance spectroscopy and polarization curves are measured under the steady open-circuit potential (OCP). The change curve of an open-circuit potential, measured in 3.5 wt% NaCl solution, with respect to immersion time, is shown in Figure 4. It can be seen from Figure 4 that the change in the OCP of the studied samples follows a similar trend. At the beginning of the experiment, the potential decreases rapidly and all of the samples slowly reach the OCP when the immersion time is 2400 s. A steady OCP is observed at length.

Figure 5 shows the potentiodynamic polarization curve of 35CrMoV steel immersed in 3.5 wt% NaCl solution after 40 min. It can be easily seen from Figure 5a,b, that the polarization curves obtained at different strain rates and deformation temperatures are similar in morphology and the I_corr_ displacement is obvious. There is no passivation and the anode region is an active dissolution of metal, which are the results of the characteristic adsorption of Cl^−^ onto the surface of the specimen, preventing the formation of the passivation film [1]. In addition, the E_corr_ shift of the polarization curve and obvious changes in the anode region can also be observed, where the anode region represents the dissolution of the matrix at high potential and the cathode region delegates the cathodic hydrogen evolution reaction associated with water reduction [12].

Tafel extrapolation is a rapid and effective method used to discuss the corrosion trend and behavior, and the corrosion rate obtained from it is mainly related to the initial surface corrosion [32]. The values of E_corr_, I_corr_, B_a_, and B_c_ obtained from the polarization curves using Tafel extrapolation means (as shown in Figure 5) are listed in Table 2. It can be observed in Figure 5 and Table 2 that I_corr_ increases with an increase of deformation temperature at the same strain rate, and I_corr_ at the deformation temperature of 850 °C is 2.4 μA/cm^2^, whereas I_corr_ at the deformation temperature of 1050 °C is 5.4 μA/cm^2^. In comparison, the values I_corr_ obtained from 850 and 1050 °C increased by 2.2 times as much. Furthermore, I_corr_ increased from 2.3 to 5.3 μA/cm^2^, which is an increase of 2.3 times, at 10 s^−1^. Therefore, it can be concluded that the higher the deformation temperature of the hot working is, the more easily corrosion occurs. Besides, at the same deformation temperature, I_corr_ decreases with an increase in strain rate and the I_corr_ at 850 °C and 10 s^−1^ is smaller than that at the strain rate of 0.01 s^−1^. It is well known that the smaller the value of I_corr_ is, the slower the corrosion reaction rate and the higher the corrosion resistance of the material. Accordingly, this shows that the corrosion resistance of 35CrMoV steel can be improved by obtaining smaller grains at high strain rates and low deformation temperatures.

#### 3.2.2. Electrochemical Impedance Spectroscopy characteristics

Generally, EIS is an effective technique for analyzing and studying the corrosion reaction between the structure of the alloy oxide film and the electrode interface, which can be used to evaluate the corrosion performance of an alloy. Figure 4 shows the curve of the OCP when measured by different specimens. For example, the stable OCP at 850 °C and 0.01 s^−1^ is about 0.6509 V_SCE_ and the EIS is recorded in the frequency range 0.1–100,000 Hz at the steady potential (0.6509 V_SCE_). The EIS of 35CrMoV steel is represented by the Nyquist plots and phase angle diagram displayed in Figure 6 and Figure 7. It can be seen from Figure 6 that the EIS of the studied 35CrMoVsteel can be greatly influenced by different deformation temperatures and strain rates. This is because the grain sizes of the specimens, acquired with the different thermal deformation parameters, showed in a comparison that the fine grains can make the grain-boundary area increase and the corrosion rate is accelerated at the early stage of corrosion, resulting in the formation of a dense oxide layer and a delaying and deepening of the corrosion. Consequently, the results show that the grain sizes obtained with the different thermal deformation parameters have a significant effect on corrosion resistance.

The radius of the capacitive reactance arc is positively correlated with its corrosion resistance [21,33,34,35], the larger the arc radius, the lower the corrosion rate of the specimens in the corresponding solution [36,37,38]. It can be observed from Figure 6 that all samples have similar semi-circular shapes, where the arc lines are similar and their diameters are different, indicating that the corrosion rate of the specimens is different but the corrosion mechanism is the same [20]. In addition, a larger arc radius can be seen at 850 °C, with strain rates of 0.01 and 10 s^−1^, which indicates that its corrosion resistance is better. At the same deformation temperature, the arc radius increases gradually with an increase of strain rate, which implies that the corrosion resistance is better (at the same strain rate), the arc radius decreases with an increase of deformation temperature, which indicates that the corrosion resistance is worse. Moreover, the arc-curve radius also reflects the impedance of the electron transfer process onto the electrode surface. It can be concluded that the hindrance can be enhanced and the corrosion rate can be decreased by a larger arc radius. As for metals, it can be noticed that the large resistance to electron transfer means that the gain and loss of electrons does not occur easily, indicating that the metals are difficult to corrode. Therefore, it is clear from Figure 6 that it is most difficult for corrosion to occur at 10 s^−1^ and 850 °C, the result with the largest arc radius, which shows that a higher strain rate and lower deformation temperature can obtain smaller grains and better corrosion resistance.

Figure 7 shows the phase angle plots of the 35CrMoV steel with different strain rates and deformation temperatures. The phase angle plots can be divided into the high-, middle-, and low-frequency regions. From Figure 7 it can be seen that the phase angle in the high-frequency region (100–1000 KHz) is small (close to zero), indicating that the impedance of this frequency range is mainly a solution impedance, and that the resistance behavior is independent of the time constant. The phase angle of the intermediate-frequency region (1–1000 Hz) reaches the maximum value, which is a typical characteristic of solubilization. The maximum value of the phase angle of the samples at 0.01 s^−1^, 850 °C and 10 s^−1^, 850 °C moves slightly towards the low-frequency direction. It was found that the capacitance of the double layer is increased [39]. Meanwhile, the presence of a wide angular front in the intermediate- and low-frequency regions implies an interaction between the two relaxation processes, as shown in Figure 7. As per the above analysis, it can be concluded that the two time constants are the reflection of the charge transfer resistance and the corrosion product resistance [40]. In addition, the impedance value, |Z|, of the low-frequency region (0.01–1 Hz) shows the impedance of corrosion reaction, which is one of the parameters used to evaluate the corrosion performance. It indicates that the larger |Z| is, the better the corrosion resistance [41,42,43]. Additionally, it can be seen from Figure 7 that the impedance value of the specimens decreases with an increase in the deformation temperature, which indicates that the corrosion resistance of the sample surface decreases. Furthermore, at the same deformation temperature, the impedance value increases with the increase of the strain rate. This shows that the corrosion resistance of the sample is stronger. Consequently, the impedance value, |Z|, of the specimen, at 10 s^−1^ and 850 °C, is observed in Figure 7, which is consistent with the results of the polarization curve, indicating the corrosion resistance is at its best and the corrosion product film on the surface is performing better.

In order to further quantitatively analyze the electrochemical corrosion behavior of the 35CrMoV steel in 3.5 wt%NaCl solution, an equivalent electrical circuit (R(Q (R (QR)) is established to fit the EIS spectra, which can well-express the corrosion mechanism of the metal interface or solution. The equivalent circuit consists of two time constants [44], as shown in Figure 8. The EIS, fitted by the ZsimpWin software (Buokamp, MA, USA), and the fitting results are shown in Table 3. The five different elements used in the equivalent circuit have different physical meanings and can be divided into three different categories. The first part, R_s_, is the solution resistance of the electrolyte, which is used to characterize the kinematic velocity of the chloride produced on the surface of the specimens in the electrolyte. In the second part, Q_f_ and R_f_ are used to imply the capacitance and resistance of the corrosion product respectively, which can reflect the diffusion of ions in the corrosion product layer and the formation of the corrosion product on the substrate surface during electrochemical corrosion. Additionally, R_f_ is an important parameter used to characterize the protective effect of the oxide layer. It can be found that a larger corrosion product resistance means better protection of the oxide layer and a greater resistance to ion movement. In the third part, Q_dl_ and R_ct_ are used to characterize the double-layer capacitance and charge transfer resistance of the reaction interface respectively, which are related to the electrochemical corrosion reaction between the sample matrix and the electrolytic solution. Furthermore, Q (in the second and third parts) is a constant phase angle element (CPE), which describes the physical quantity when the parameter of interface capacitance, C, deviates due to dispersion effects. In general, the reactions in the second and third parts occur in the middle- and low-frequency regions of EIS, which are related to the corrosion resistance of the specimens. The corrosion resistance of the samples is evaluated by combining the corrosion product resistance R_f_ with the charge transfer resistance R_ct_ (R_corr_ = R_f_ + R_ct_) [1,33,35,40]. Generally, the corrosion resistance enhances as R_corr_ increases. The polarization resistances (R_corr_) of the two groups of all of the samples studied are shown in Figure 9. It can be seen from Figure 9 that R_corr_ decreases with an increase of 35CrMoV steel thermal deformation temperature at the same strain rate, indicating that the corrosion rate is accelerated. This is because an increase in the 35CrMoV hot deformation temperature is conducive to grain growth, resulting in a coarse grain size and lower corrosion resistance. At the same temperature, R_corr_ increases with an increase in the strain rate and, more importantly, the highest R_corr_ appears at 10 s^−1^ and 850 °C, indicating that the fine grains obtained at high strain rates and low deformation temperatures can make the oxide layer denser and improve the corrosion resistance.

### 3.3. Corrosion Products and Corrosion Mechanism Analysis

In order to further discuss the effect of the grain sizes, obtained under different deformation conditions, on the corrosion resistance of the specimens, the two- and three-dimensional morphology of the specimens surface is observed, on the basis of the potentiodynamic polarization experiment. The two-dimensional corrosion morphology of four typical specimens with deformation temperatures of 850 °C and 1050 °C, at the strain rates of 0.01 s^−1^ and 10 s^−1^ respectively, immersed in 3.5 wt% NaCl solution for 1 h and placed for 3 days, is depicted in Figure 10. It can be observed from Figure 10 that the surface of the specimens display local corrosion, where the corrosion holes are in the direction of depth. The black and yellowish-brown areas are corrosion areas, and the bright white areas are un-corroded areas. The corrosion pits on the surface are covered with rust spots and corrosion products, as well as corrosion pits of different sizes. At the same strain rate, with an increase in the 35CrMoV hot deformation temperatures, the local corrosion gradually expands and connects to a larger area of corrosion. Furthermore, the corrosion area becomes smaller and the corrosion degree weakens with an increase of the strain rate at the same deformation temperature. The three-dimensional morphology of the a′, b′, c′, and d′ (see Figure 10) regions under the CLSM is shown in Figure 11. The size of the etch pits are measured, along the Y direction of the specimens, with the CLSM. The etch pit depths of (a′), (b′), (c′), and (d′) in Figure 10 are 17.17, 49.55, 13.64, and 30.45 μm, respectively. At the same strain rates (0.01 s^−1^ and 10 s^−1^), the surface is locally corroded and there are corrosion pits at the deformation temperatures of 850 °C and 1050 °C, but the corrosion pit at the deformation temperature of 1050 °C is large and deep, which indicates that the corrosion resistance is poor at higher deformation temperatures. This is as the dynamic re-crystallization rate of the sample steel increases at high deformation temperatures, and the re-crystallized large grains continue to swallow small grains and grow. Additionally, at the same deformation temperature, the corrosion surface is smoother and the corrosion pit decreases with an increase of strain rate, which is consistent with the characterization in Figure 10. This is because the structure of the double layer at the interface between the matrix iron and the corrosion product film is prone to preferential adsorption of Cl^−^, which results in the surface deposition of Cl^−^ fluids, which then combine with cations on the oxidation film to form soluble chloride and corrosion pits. However, the PH value of the corrosion pit decreases with the continuous hydrolysis of chloride, as well as the dissolution of the anode metal of the corrosion pit, and the external Cl^−^ is invaded into the corrosion pit through the corrosion product film, which makes the corrosion proceed further. In this cycle, the deepening and expansion of the corrosion pit depth are the results of the corrosion catalysis of Cl^−^.

Figure 12 and Figure 13 show the surface morphology of the specimens after polarization in 3.5 wt% NaCl solution. It can be observed from Figures (a), (b), and (c) in Figure 12 and Figure 13 that there are cracks and platelet crystal-like corrosion product films on the surface of the specimens. At the same strain rate, the surface corrosion product-coverage area increases and the corrosion product accumulates with an increase of the deformation temperature. However, at the same deformation temperature, the accumulation of corrosion products decreases because of an increase in the strain rate, indicating that the corrosion degree is weakened. The cracks on the corroded surface are displayed in Figure 12 and in Figure 13a′–c′, and are due to dehydration after the sample immersion test [45]. The corrosion cracks are formed at the initial stage of corrosion which, with an increase of temperature, become wider, deeper, and looser at the same strain rate, and the corrosion products fall off slightly, indicating that corrosion is deepening. However, at the same deformation temperature, the corrosion cracks on the surface of the specimens become narrower, shallower, denser, and less detached as the strain rate increases, indicating that the corrosion rate is weakened. The platelet-like corrosion products (Figure 12 and Figure 13d′–f′) generated on the surface are due to the deepening of the surface corrosion, resulting in rust being generated and the corrosion products accumulating. In other words, with an increase of deformation temperature, the gap in platelet corrosion products becomes wider and looser at the same strain rate, which is more likely to cause oxidative corrosion on the surface, favorable to the further invasion of the corrosive medium which accelerates the corrosion. Furthermore, an increase of the deformation temperature is beneficial to grain growth, which then affects the corrosion rate at the same strain rate. Additionally, the cracks are narrower and the platelet-corrosion product film is densified and uniform at the same deformation temperature, due to strain rate increases. The deformation process becomes rapid, the dynamic recovery rate decreases, and the re-crystallized grains have not yet fully grown, resulting in smaller grains.

The results show that the impedance modulus at the EIS (Figure 6) is the appropriate parameter to characterize the protective performance of the corrosion product film [46,47,48]. From Figure 6a, it can be seen that no scattering fluctuations were observed in all of specimens at 0.01 Hz, which indicates that the corrosion product film has great stability and can effectively protect the matrix from Cl^−^ in 3.5 wt% NaCl solution, as well as preventing further corrosion [41].

The surface of the specimens was scanned by EDS and two groups of specimens, with different strain rates at deformation temperature 850 °C, were selected for analysis. Figure 14 shows the corrosion morphology and EDS analysis of the surface of the 35CrMoV steel at deformation temperature 850 °C at 0.01 and 10 s^−1^. Both of the specimens can be observed to have corrosion cracks (Figure 14(a1,b1)) and platelet-like products (Figure 14(a2,b2)) [20], produced on the surface of the 35CrMoV steel. Additionally, there is an obvious accumulation of massive corrosion products, and the thickness of the corrosion layer is uneven. The corrosion cracks (Figure 14(a1,b1)) of the specimens all contain O, Fe, C, and Cr, as revealed by EDS, and the oxygen content on the surface decreases obviously with the increase of the strain rate. Meanwhile, the surface of the platelet-like product (Figure 14(a2,b2)) is mainly composed of O, Fe, C, and Cr, and the oxygen content in 0.01 s^−1^ is obviously higher than that in 10 s^−1^, indicating that the corrosion product is mainly oxide. Furthermore, the content of Cr in 10 s^−1^ obviously increased and the presence of Cr elements on the surface of the corrosion product film can effectively prevent the deep erosion of Cl^−^, improve the chemical stability of the corrosion product film, and prevent further corrosion.

X-ray diffraction analysis was conducted to determine the types of corrosion products (see Figure 15a) and the matrix (see Figure 15b) after corrosion on the surface of the samples. The energy spectrum analysis results of the 35CrMoV steel matrix and corrosion products after corrosion are displayed in Figure 15. It can be seen from Figure 15 that the surface of the matrix after heat treatment mainly contains Fe, and some studies have shown that its structure is tempered sorbite [49,50]. In summary, the corrosion products on the surface after corrosion are mainly FeOOH. Combined with the EDS analysis of Figure 14, it can be concluded that O, Fe, and C are the main components of the corrosion product. Therefore, it can be inferred that the cathode in the corrosion solution is depolarized by oxygen, due to adequate oxygen supply, and the surface is covered by a thin water film.

In the early stages of corrosion, Cl^−^ adsorbed on the metal surface play an erosion role and Fe is dissolved from the anode to form Fe^2+^, combined with Cl^−^ to form FeCl_2_·4H_2_O [51], and further decomposed to form Fe(OH)_2_. However, Fe(OH)_2_ is unstable and decomposed into FeO or oxidized to FeOOH by the O_2_ dissolved in the water film (see Equations (1) and (2)) [1,52,53,54]. In addition, some studies have shown that Fe (OH)_2_ can continue to be oxidized and dehydrated to form Fe_2_O_3_ and Fe_3_O_4_, which are both dense and difficult to decompose. Accordingly, the reduction in corrosion rate is the result of the Fe_2_O_3_ and Fe_3_O_4_ hindering the diffusion of oxygen and Cl^−^, which exist in the inner rust layer [55]. According to the Evans theory, the corrosion products will partially dissolve due to the strong erosive Cl^−^ in the solution. Many cracks will occur in the rust layer, which can provide a gap between the corrosion media (such as O_2_) and spread, causing the corrosion to continue.
(1)$${\mathrm{Fe}}^{+}{\;+\;2\mathrm{OH}}^{-}\to {\mathrm{Fe}(\mathrm{OH})}_{2}$$
(2)$$4{\mathrm{Fe}(\mathrm{OH})}_{2}{\;+\;O}_{2}\to {4\mathrm{FeOOH}\;+\;2H}_{2}O$$

Grain refinement leads to an increase in grain boundary area per unit volume at lower deformation temperatures and higher strain rates. It is well known that the potential at grain boundaries is lower than inside the grain, which is as the crystal defect density at the grain boundary is large. In addition, the existence of a potential difference (PD) constitutes the grain–grain boundary corrosion micro-battery. As a result, the grain boundary (serving as an anode) has priority in corrosion. In the same corrosion environment, there is a certain potential difference between the grain and the grain boundary. The local anodic corrosion current density experienced by the grain boundary is relatively small when corroded, so it will not intrude very deep holes and cracks, and it will give the rust layer a good compactness. The grain size directly affects the corrosion rate at the beginning of corrosion. In contrast to the coarse-grain rate, the fine-grain corrosion rate is faster, which is mainly due to the fact that grain refinement increases the grain boundary area, and the grain boundary is increased in the microstructure. Besides this, the grain boundary is a high-active zone, which may make it vulnerable to pitting corrosion, leading to surface unevenness and increases in the anode surface area, and causing the electrochemical reaction to proceed rapidly [56]. This would result in a larger corrosion area, heavier corrosion, and faster anodic dissolution, which is conducive to the rapid formation of a protective inner rust layer. Thus, a finer-grain of the specimens is more favorable to the flatness of the rust layer/matrix interface in the specimens and a more-uniform dissolved matrix. At the same time, the rust particles are refined and there is a reduction of the cracks and pores in the rust layer, which can cause the matrix and rust layer to combine firmly. Furthermore, the rust layer becomes thicker, denser, and more stable and the alloying elements on the matrix surface play an important role in the development of corrosion [57,58], resulting in an increase in the self-corrosion potential of the sample while enhancing the corrosion resistance and reducing the dissolution rate of the anode. In addition, the current carrying density of the oxide film was reduced and the dissolution of the oxide film slowed down due to grain refinement [59,60], accordingly improving the corrosion resistance [61].

## 4. Conclusions

The microstructure of 35 CrMoV steel under different hot deformation conditions and the short-term corrosion behavior in 3.5 wt% NaCl solution was discussed, and some important conclusions, as follows, were obtained:

(1) Deformation temperatures and strain rates have an important influence on the microscopic structure. At the same strain rate, the grain size increased with an increase in deformation temperature and decreased with an increase in strain rate at the same strain temperature. Amongst all of the hot deformation parameters studied, the grain size at a strain rate of 10 s^−1^ and a deformation temperature of 850 °C was the smallest (10.7 μm), indicating that a finer-grain structure can be obtained at lower deformation temperatures and higher strain rates. 

(2) The difference in grain sizes had a significant effect on the corrosion resistance of 35CrMoV steel. At the same strain rate, the grain size augmented with an increase in temperature and the corrosion resistance decreased with an increase in the grain size, within a certain corrosion time. Furthermore, the corrosion resistance was the worst when the grain size was the largest (15.4 μm). At the same deformation temperature, the higher the strain rate, the smaller the grain size, and therefore, the better the corrosion resistance of the specimens within a certain period of time. The smallest grain size (10.6 μm) was observed at a strain rate of 10 s^−1^ and a deformation temperature of 850 °C, resulting in the lowest corrosion rate, which indicates low-deformation temperature and high-strain rate can achieve the effect of refining grains and improving corrosion resistance. However, the long-term corrosion trends of these materials need to be studied further.

(3) The differences in the corrosion resistance of 35CrMoV steel were related to the grain boundary change after grain refinement. Grain refinement increased the grain boundary area and the grain boundary, which accelerated the formation of a protective oxide film at the initial stage of corrosion, improved the stability of the oxide film, and prevented the corrosion from proceeding deeply, thereby improving corrosion resistance.

## Figures and Tables

**Figure 1 materials-12-01455-f001:**
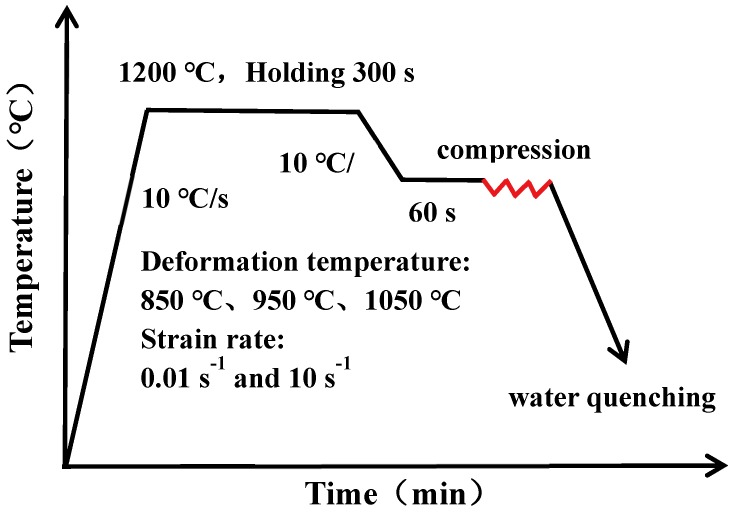
Schematic representation of hot compression test.

**Figure 2 materials-12-01455-f002:**
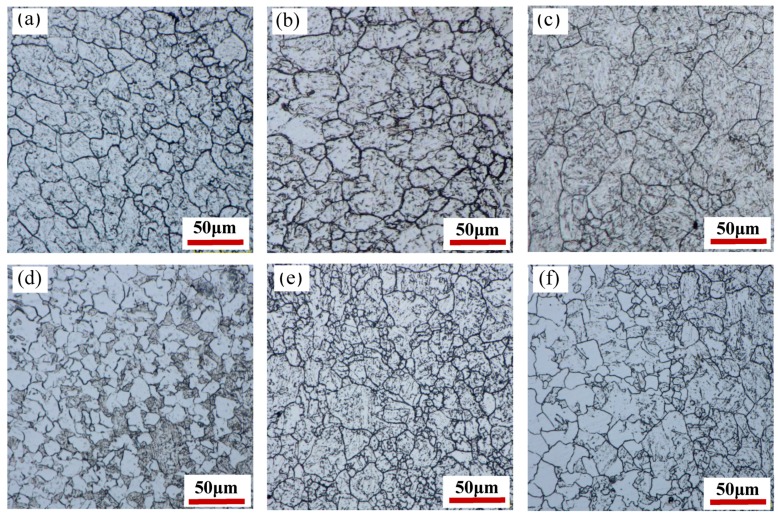
Microstructure of 35CrMoV steel at different deformation temperatures and strain rates: (**a**) 0.01 s^−1^, 850 °C; (**b**) 0.01 s^−1^, 950 °C; (**c**) 0.01 s^−1^, 1050 °C; (**d**) 10 s^−1^, 850 °C; (**e**) 10 s^−1^, 950 °C; (**f**) 10 s^−1^, 1050 °C.

**Figure 3 materials-12-01455-f003:**
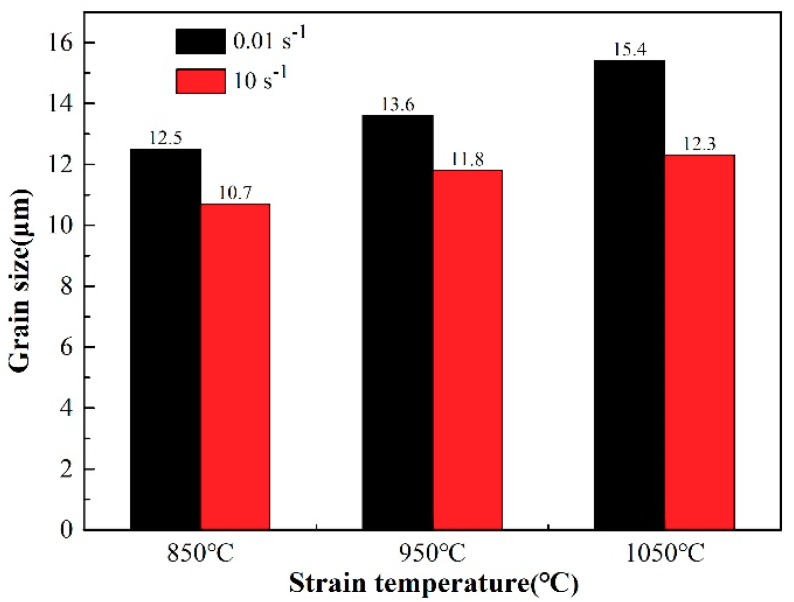
Grain size of 35CrMoV steel at different deformation temperatures and strain rates.

**Figure 4 materials-12-01455-f004:**
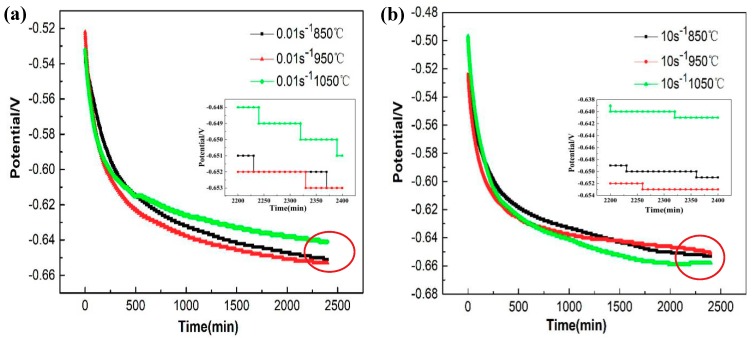
The open-circuit potential (OCP) curves of 35CrMoV steel in 3.5 wt% NaCl solution: (**a**) 0.01 s^−1^, 850–1050 °C; (**b**) 10 s^−1^, 850–1050 °C.

**Figure 5 materials-12-01455-f005:**
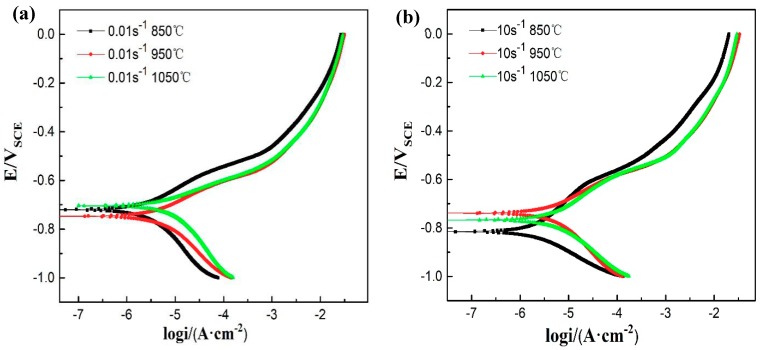
Polarization curves of 35CrMoV steel in 3.5 wt% NaCl solution: (**a**) 0.01 s^−1^, 850–1050 °C; and (**b**) 10 s^−1^, 850–1050 °C.

**Figure 6 materials-12-01455-f006:**
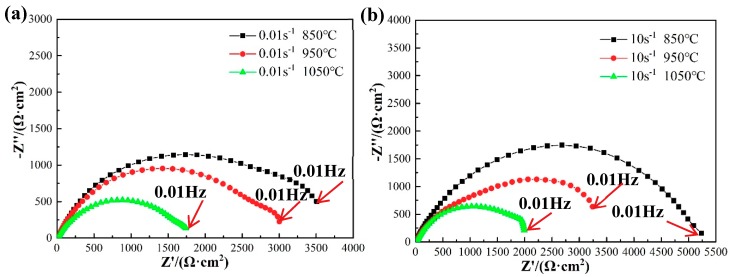
Nyquist plots of the 35CrMoV steel in 3.5 wt% NaCl solution: (**a**) 0.01 s^−1^, 850–1050 °C; (**b**) 10 s^−1^, 850–1050 °C.

**Figure 7 materials-12-01455-f007:**
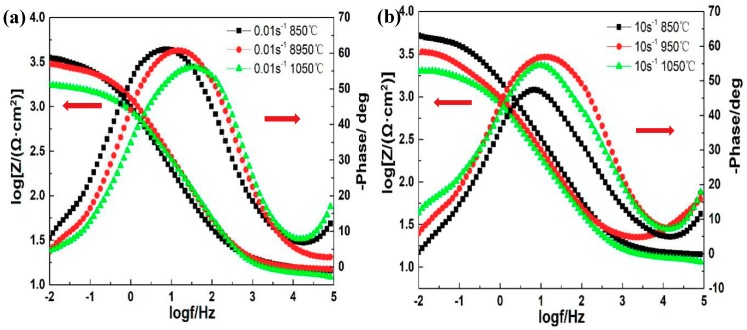
The Bode plots of 35CrMoV steel in 3.5 wt% NaCl solution: (**a**) 0.01 s^−1^, 850–1050 °C; (**b**) 10 s^−1^, 850–1050 °C.

**Figure 8 materials-12-01455-f008:**
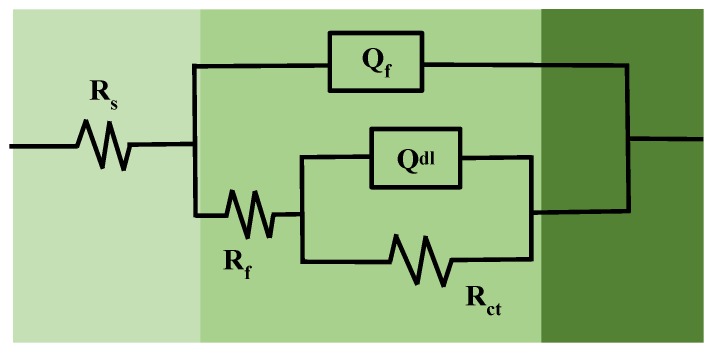
The equivalent electrical circuit of the 35CrMoV steel in 3.5 wt% NaCl solution.

**Figure 9 materials-12-01455-f009:**
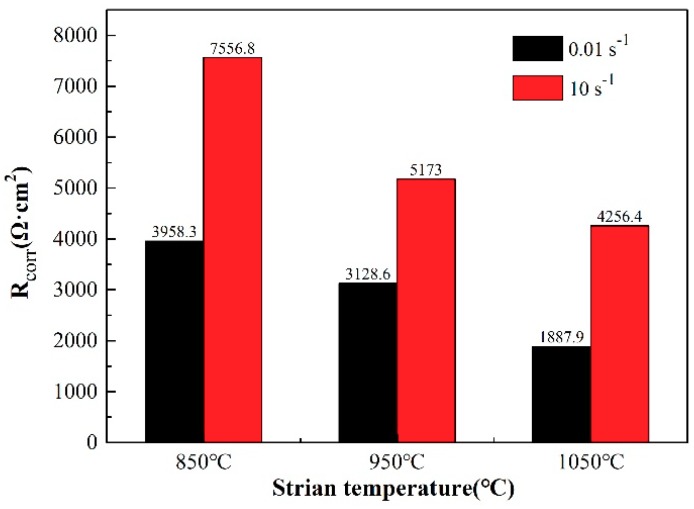
The corrosion rate of the 35CrMoV steel in 3.5 wt% NaCl solution.

**Figure 10 materials-12-01455-f010:**
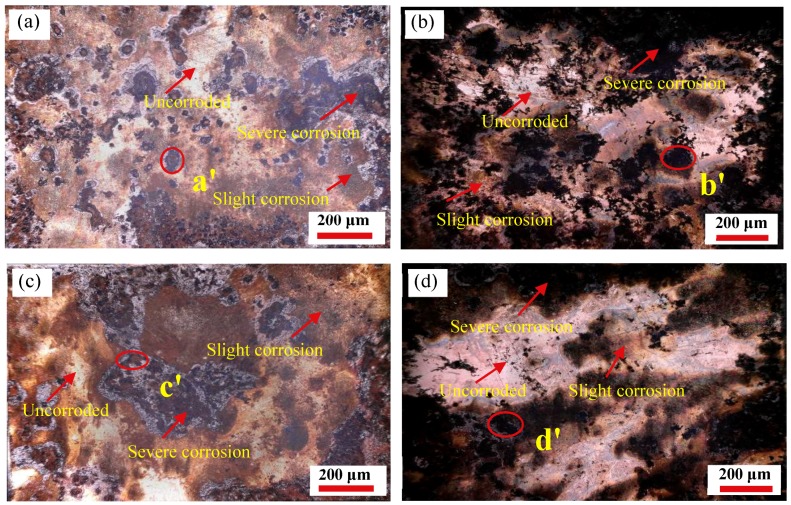
Corrosion morphology of the 35CrMoV steel: (**a**) 0.01 s^−1^, 850 °C; (**b**) 0.01 s^−1^, 1050 °C; (**c**) 10 s^−1^, 850 °C; (**d**) 10 s^−1^, 1050 °C.

**Figure 11 materials-12-01455-f011:**
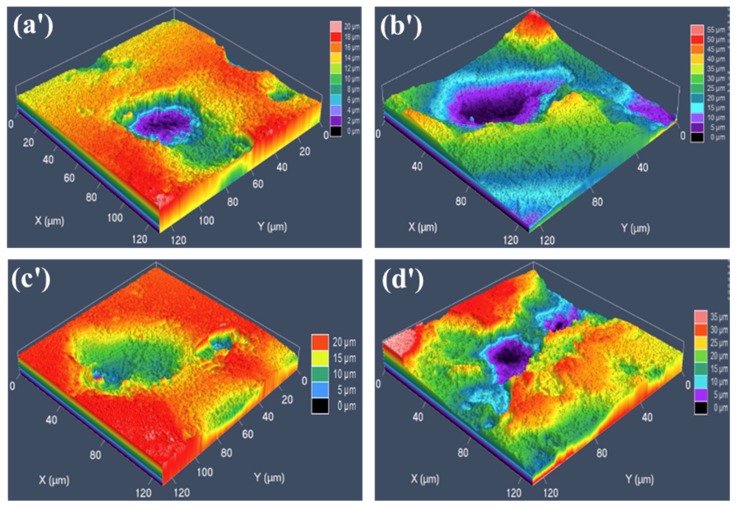
Three-dimensional corrosion morphology of the 35CrMoV steel: (**a′**) 0.01 s^−1^, 850 °C; (**b′**) 0.01 s^−1^, 1050 °C; (**c′**) 10 s^−1^, 850 °C; (**d′**) 10 s^−1^, 1050 °C.

**Figure 12 materials-12-01455-f012:**
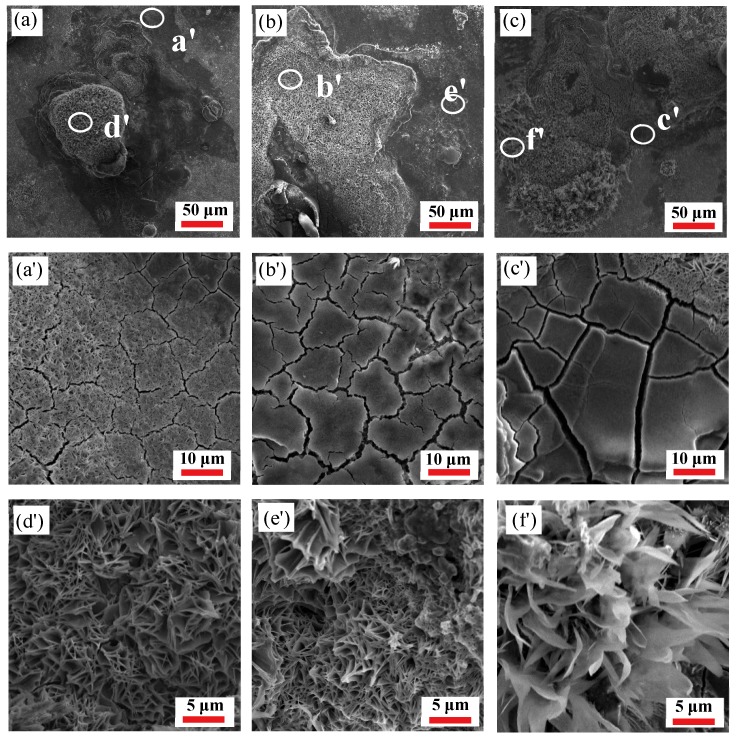
Scanning electron microscopy (SEM) corrosion morphology of the 35CrMoV steel: (**a**,**a′**,**d′**) 0.01 s^−1^, 850 °C; (**b**,**b′**,**e′**) 0.01 s^−1^, 950 °C; and (**c**,**c′**,**f′**) 0.01 s^−1^, 1050 °C.

**Figure 13 materials-12-01455-f013:**
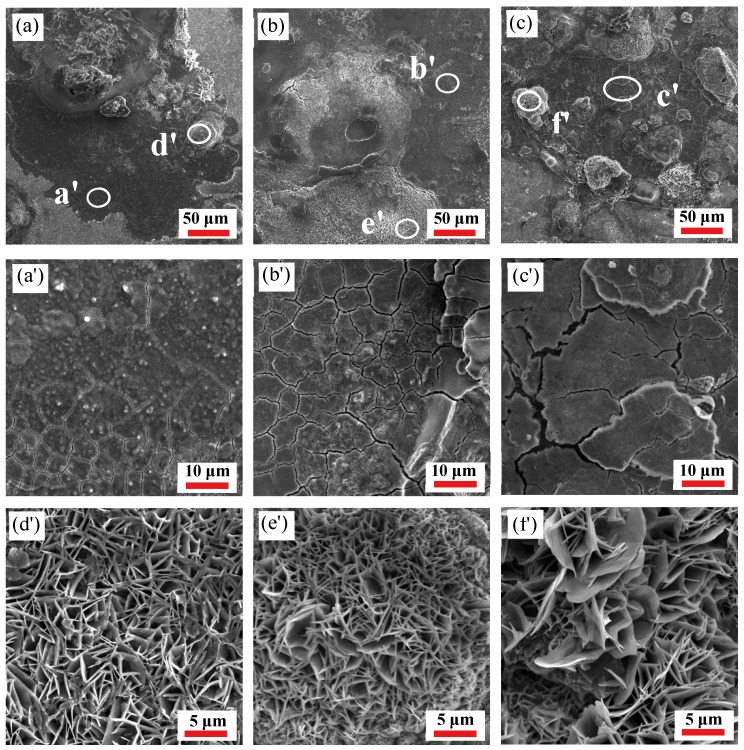
SEM corrosion morphology of the 35CrMoV steel: (**a**,**a′**,**d′**) 10 s^−1^, 850 °C; (**b**,**b′**,**e′**) 10 s^−1^, 950 °C; and (**c**,**c′**,**f′**) 10 s^−1^, 1050 °C.

**Figure 14 materials-12-01455-f014:**
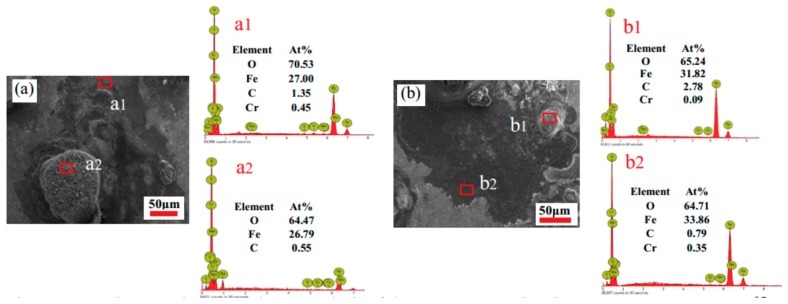
Corrosion morphology and energy disperse spectroscopy (EDS) analysis of the 35CrMoV steel at the same temperature (850 °C) and at different strain rates: (**a**) 0.01 s^−1^; and (**b**) 10 s^−1^.

**Figure 15 materials-12-01455-f015:**
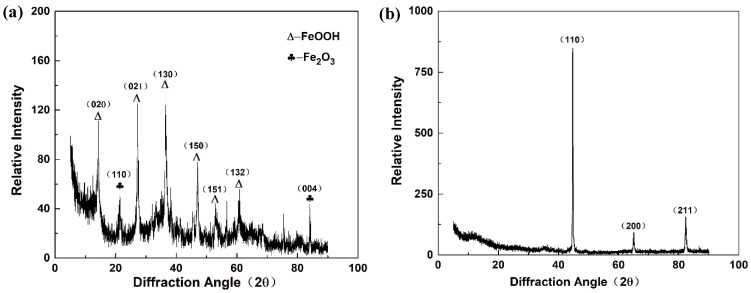
X-ray diffraction (XRD) analysis of the 35CrMoV steel: (**a**) corrosion products; and (**b**) matrix.

**Table 1 materials-12-01455-t001:** Composition of GB 35CrMoV steel (wt%).

Chemical Composition	C	Si	Mn	Mo	S	P	Cr	V	Fe
Measured	0.36	0.23	0.29	0.26	0.009	0.035	1.19	0.14	Bal

**Table 2 materials-12-01455-t002:** Electrochemical test results of 35CrMoV steel.

Sample		E_corr_/V_SCE_	R_p_/(Ω·cm^2^)	I_corr_/(μA/cm^2^)	B_a_/(mV/dec)	B_c_/(mV/dec)
0.01 s^−1^	850 °C	−0.720	12,290.8	2.441	87.73	57.20
	950 °C	−0.747	6418.7	4.143	105.11	58.40
	1050 °C	−0.704	4083.5	5.393	145.06	52.38
10 s^−1^	850 °C	−0.815	12,927.8	2.332	55.69	88.50
	950 °C	−0.739	6795.4	4.354	90.24	56.71
	1050 °C	−0.767	6072.5	5.256	69.64	66.59

**Table 3 materials-12-01455-t003:** Spectra fitting results of experimental 35CrMoV steel in 3.5 wt% NaCl solution.

Sample		R_s_ (Ω·cm^2^)	Y_f_ (Ω^−1^·cm^−2^·s^n^)	n	R_f_ (Ω·cm^2^)	Y_dl_ (Ω^−1^·cm^−2^·s^n^)	n	R_ct_ (Ω·cm^2^)
0.01 s^−1^	850 °C	15.66	2.316 × 10^−4^	0.7438	3469.00	2.081 × 10^−2^	1.000	489.3
	950 °C	15.21	1.544 × 10^−4^	0.7619	2768.00	1.585 × 10^−2^	0.9732	360.6
	1050 °C	12.84	1.637 × 10^−4^	0.7451	961.30	5.139 × 10^−4^	0.3708	926.6
10 s^−1^	850 °C	17.85	3.779 × 10^−4^	0.6669	26.83	1.88 × 10^−4^	0.8759	7530.0
	950 °C	14.42	1.236 × 10^−4^	0.7733	3713.00	3.016 × 10^−4^	0.8368	1460.0
	1050 °C	36.56	4.464 × 10^−5^	1.0000	189.40	3.157 × 10^−4^	0.5485	4067.0

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
