# Peer review of "Effect of Hot Deformation Process Parameters on Microstructure and Corrosion Behavior of 35CrMoV Steel"

_materials, 2019, doi:10.3390/ma12091455_

Round 1

Reviewer 1 Report

Well organized study with relevant findings. I note that despite their relevance for developing science for corrosion initiation, electrochemical tests should not be used to extrapolate long-term material performance. Mechanisms, conditions etc. have been extensively shown to change with time. I suggest authors make this clear in the conclusions. Alternatively, field trials should be performed in order to validate the conclusions.

English grammar, syntax and vocabulary has to be seriously revisited.

Author Response

Thank you very much for your valuable comments on this article. I will revise it in the manuscript as soon as possible.

Here are some of my comments on your question

The conclusions obtained in this paper are the initial corrosion conditions under different thermal deformation parameters under corrosive conditions (simulated marine environment 3.5wt% NaCl solution). The main influences of the grain size obtained under this process on the corrosion of 35CrMo steel are studied. Corrosion mechanism and situation, we will conduct in-depth research and discussion in the later work to obtain more comprehensive and in-depth research results.

Reviewer 2 Report

This paper describes the effect of temperature and strain rate during the hot deformation process on the microstructure and corrosion resistance of 35CrMoV steel. A clear correlation between grain size and corrosion resistance is established using metallography and electrochemical characterisation, although the effect is relatively small (a factor of two decrease in corrosion resistance at the higher temperatures and only a minor effect of strain rate).

I am satisfied that the paper represents a worthy contribution to the literature on this topic. However, some aspects of the manuscript need to be addressed before it is suitable for publication.

My main concern is over the description of ‘pitting’ during the post mortem examination of the specimens. This is in contradiction to the electrochemical characterisation, which shows no passive behaviour, and needs to be addressed by the authors in the discussion of the results.

The manuscript would also benefit from thorough revision to correct a large number of spelling and grammatical errors. The most important are listed below but there are many others.

Section 2.1: a figure showing a schematic diagram of the hot deformation process would be useful here as it is difficult to picture the setup from the description given in the text.

Line 132: What is ‘seagull shampoo’? Chemical composition needs to be specified.

Line 150: replace ‘CE’ with ‘SCE’

Figure 4: why is the resolution of the potentiostat only 1 mV?

Line 435: replace ‘Fe.4H2O’ with ‘FeCl2.4H2O’

Figure captions: the first word of some of the captions is missing

Author Response

Thank you very much for your valuable comments on the manuscript. Here are some explanations to your questions, 

Pitting:The concept description of the pit is ambiguous, and the expression is ambiguous. The two-dimensional corrosion morphology observed by the ultra-depth microscope, the surface is unevenly corroded, and there are corrosion pits of different sizes, which will corrode during the writing and understanding. The pit mistakenly thought of the pitting pit, causing ambiguity, I will modify it in the manuscript.

Line 132: Seagull shampoo is dodecyl benzenesulfonic acid, sodium salt,which composed of C, H, C, Na, O, S and other elements, also known as sodium tetrapropylene benzene sulfonate, white or yellowish powdered or flake solids.

Line 435: replace ‘Fe.4H2O’ with ‘FeCl2.4H2O’

Figure 4: The three-electrode system of ordinary electrochemical workstation (Chi660e,CH-Hua, Shanghai, China) is used in this electrochemical corrosion test. The resolution of 1mv in the small diagram is to obtain stable open-circuit potential after 2400s better reaction. It is shown that the polarization curves and electrochemical impedance spectra are carried out at a stable potential.

Line 435: replace ‘Fe.4H2O’ with ‘FeCl2.4H2O’

Figure description: some headings have been added to the first word.

Reviewer 3 Report

The microstructure and corrosion behavior 35CrMoV steel under the different hoy deformation conditions are discussed and obtained some important conclusions as:

- Deformation temperatures and strain rates have an important influence on microscopic structure.

- The different in grain sizes has a significant effect on the corroison resistance of 35CrMoV steel.

- The difference of corrosion resistance of 35CrMoV steel is related to the grain boundary change after grain refinement.

It's a very interesting paper.

The following changes must be made:

You should review all text, unifying how to write the data. For example: 850 ºC or 850ºC, 0.01 s-1 or 0.01s-1.

Figures 3, 10 and 12 should include foot is on the same page.

403. Figure 13 is erroneously designated.
Appointments 21 and 34 are the same. Renumber all dating from the 34th.

Author Response

Thank you very much for your valuable comments on this article. I will revise it in the manuscript as soon as possible. Thank you again for your contribution to the article.